# Use Case Based Blended Teaching of IIoT Cybersecurity in the Industry 4.0 Era

**Tiago M. Fernández-Caramés** [1,2,*] and **Paula Fraga-Lamas** [1,2,*]

[1] Department of Computer Engineering, Faculty of Computer Science, Universidade da Coruña, 15071 A Coruña, Spain
[2] Centro de investigación CITIC, Universidade da Coruña, 15071 A Coruña, Spain
* Correspondence: tiago.fernandez@udc.es (T.M.F.-C.); paula.fraga@udc.es (P.F.-L.); Tel.: +34-981167000 (ext. 6051) (P.F.-L.)

**Abstract:** Industry 4.0 and Industrial Internet of Things (IIoT) are paradigms that are driving current industrial revolution by connecting to the Internet industrial machinery, management tools or products so as to control and gather data about them. The problem is that many IIoT/Industry 4.0 devices have been connected to the Internet without considering the implementation of proper security measures, thus existing many examples of misconfigured or weakly protected devices. Securing such systems requires very specific skills, which, unfortunately, are not taught extensively in engineering schools. This article details how Industry 4.0 and IIoT cybersecurity can be learned through practical use cases, making use of a methodology that allows for carrying out audits to students that have no previous experience in IIoT or industrial cybersecurity. The described teaching approach is blended and has been imparted at the University of A Coruña (Spain) during the last years, even during the first semester of 2020, when the university was closed due to the COVID-19 pandemic lockdown. Such an approach is supported by online tools like Shodan, which ease the detection of vulnerable IIoT devices. The feedback results provided by the students show that they consider useful the proposed methodology, which allowed them to find that 13% of the IIoT/Industry 4.0 systems they analyzed could be accessed really easily. In addition, the obtained teaching results indicate that the established course learning outcomes are accomplished. Therefore, this article provides useful guidelines for teaching industrial cybersecurity and thus train the next generation of security researchers and developers.

**Keywords:** IIoT; Industry 4.0; cybersecurity; Shodan; blended teaching; use-case based teaching; use case based learning; security audit

## 1. Introduction

Industry 4.0 is a concept that sets the foundations for evolving traditional factories to become smarter. Such an evolution is aimed at optimizing process efficiency and flexibility through the application of the latest technologies [1,2]. Industry 4.0 foundations are similar to the ones suggested by other paradigms and initiatives, like Industrial Internet of Things (IIoT) [3], Internet Plus [4] or Made in China 2025 [5]. The most relevant technologies involved in Industry 4.0 are depicted in Figure 1 and include, among others, Augmented/Mixed/Virtual Reality (AR/MR/VR) [6,7], integration systems [8], simulation software [9], additive manufacturing [10], Industrial Cyber-Physical Systems (ICPS) [11], Big Data [12], industrial autonomous machines like robots [13], Automated Guided Vehicles (AGVs) [14] and Unmanned Aerial Vehicles (UAVs) [15]. In addition, other recent technologies like blockchain [16] have been considered for creating autonomous decentralized industrial systems. Such enabling technologies allow for creating highly

connected factories that often rely on cloud-centered architectures and, increasingly, on novel decentralized fog and edge computing architectures [17,18].

The dashed lines that connected the different Industry 4.0 technologies in Figure 1 indicate the other technologies they rely on. For instance, Cyber-Physical Systems (CPSs) rely on IIoT devices and Big Data analysis in order to provide true Industry 4.0 applications. Such dashed lines in Figure 1 show that the different Industry 4.0 technologies have two main dependencies: cybersecurity and IIoT. The former is related to the need for providing robust and reliable factories, while the second one demonstrates the increasing dependence on IIoT devices like Programmable Logic Controllers (PLCs), industrial machinery, tools or communications gateways, which can be connected to internal LANs or to the Internet so as to monitor and to interact with them. Thus, IIoT devices are essential for industrial fields like agriculture [19,20], healthcare [21] or heavy industries [22].

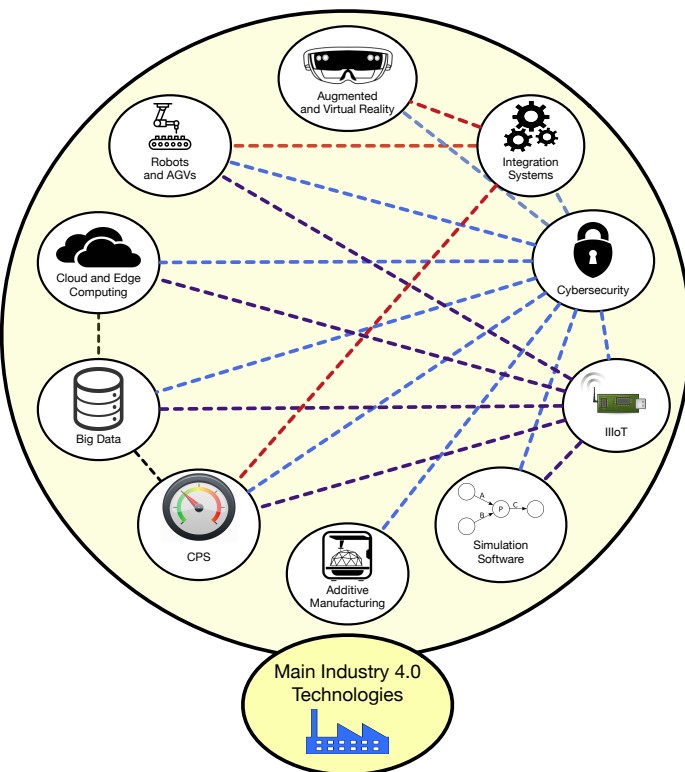

**Figure 1.** Main Industry 4.0 technologies and their dependencies.

The problem is that many IIoT devices are in practice insecure due to weak security implementations or to the lack of a proper security configuration. Part of such an insecurity is related to certain hardware design requirements (e.g., many IIoT devices rely on batteries or are constrained in terms of memory or computational power), which prevent them from implementing complex security mechanisms (e.g., Rivest–Shamir–Adleman (RSA) [23] or Elliptic Curve Cryptography (ECC) [24]) that demand a relevant amount of resources [25]. In addition, IIoT firmware can have software bugs that in many cases are not easy to fix (e.g., they may involve replacing the hardware or require performing specific firmware update procedures).

The previously mentioned security problems are essentially related to neglecting cybersecurity from the IIoT device design stage. In fact, security is often neglected by many IT university degrees, so, in general, graduates do not receive an intensive training on the diverse fields where security is key. This is the case of the industrial field, which, despite being critical, is barely studied from a cybersecurity point of view.

In addition, there is a growing interest in blended or hybrid teaching models. Such models include a mix of traditional face-to-face instruction and ordinary classroom learning with online and

remote learning and projects to engage students in the learning process. Specifically, flipped classroom is a pedagogical blended learning approach focused on student engagement and active learning. The mentioned interest in blended teaching grew significantly during the first semester of 2020 due to the COVID-19 pandemic, since blended models bring universities the opportunity to compensate part of the educational needs during a lockdown and allow for adapting to limited classroom space while keeping convenient instructional and social interactions (e.g., practical lab assignments).

To tackle the previously mentioned issues, this article includes the following novel contributions:

- A practical use case-based blended teaching methodology for Industry 4.0 and IIoT cybersecurity is proposed. Such a methodology is supported by the use of an online tool like Shodan [26], which is able to accelerate significantly the IIoT device reconnaissance stage and challenges students with real use cases.
- A theoretical and empirical approach to Industry 4.0/IIoT security is provided to help educators to replicate the learning outcomes expected from the course, which have been successfully put in practice in seminars and master courses at University of A Coruña (Spain). For such a purpose, this article provides multiple practical use cases together with useful guidelines to detect and prevent Shodan-based attacks and to help students to understand the impact and security implications involved in the deployment of the latest Industry 4.0 technologies.
- The course teaching results obtained during its four editions are presented and analyzed in order to validate the proposed flipped classroom based approach.

The rest of this article is structured as follows. Section 2 analyzes the latest literature on cybersecurity teaching and IIoT/Industry 4.0 security learning. Section 3 describes the proposed industrial cybersecurity course structure. Section 4 details the basics of Shodan and indicates how to automate Shodan searches through its Application Programming Interfaces (APIs). In addition, such a Section details the practical lab teaching methodology. Section 5 summarizes the obtained teaching results, while Section 6 is devoted to the conclusions. Finally, Appendix A enumerates multiple practical use cases of Shodan searches that allow for finding IIoT/Industry 4.0 devices.

## 2. State of the Art

### 2.1. IIoT and Industry 4.0 Cybersecurity Teaching

Although industry has been increasingly targeted by cybersecurity attacks, its security is not widely studied in IT university programs. In fact, few university programs have incorporated explicitly cybersecurity topics [27], despite considering such a field as a core competency of future IT engineers [28] and recognizing the existence of an increasing gap in qualified cybersecurity professionals [29].

Moreover, in the first semester of 2020 the COVID-19 pandemic caused a massive impact in the higher education system, forcing institutions to put in practice emergency online and blended models, and to implement measures to reinforce and guarantee social distancing (e.g., strategies to compensate for limited classroom space). Unfortunately, as of writing, the vast majority of cybersecurity courses follow the traditional lecture-and-lab structure, but some educators have proposed to make use of the latest technologies to ease remote learning and provide more realistic experiences.

Thus, several authors described cloud-based systems that allow for implementing online cybersecurity courses. For example, Salah et al. [30] propose the use of a virtual classroom system based on Amazon Web Services for teaching cybersecurity, while other authors propose the use of an off-campus network laboratory with low-cost platforms as lightweight honeypots [31]. Similarly, Tunc et al. [32] suggest easing cybersecurity experiments by providing online tools through a concept that the authors call Cybersecurity Lab as a Service (CLaaS).

In the specific case of IIoT cybersecurity teaching, just a few works can be found in the literature. One of them is proposed by Dawson et al. [33], who present a framework for creating a virtual cybersecurity

lab able to incorporate IIoT and embedded systems. Another interesting work was carried out by Wiesen et al. [34], who focused on describing a course on hardware reverse engineering that analyzes the practical security of devices like Field-Programmable Gate Arrays (FPGAs), which have been used by Industrial Control Systems (ICSs). In the case of Thiriet et al. [35], they described a program of a course on cybersecurity oriented towards cyber-physical systems.

To engage students on industrial cybersecurity, some researchers proposed to gamify learning through case-based approaches [36], which can be performed together with penetration testing competitions [37], capture-the-flag games [38,39], challenge-based platforms [40], defenders versus attackers competitions [41] or secure development contests [42]. For example, in [43] the authors added an educational game to teach students ICS cybersecurity to their flipped learning methodology [44]). Another good example is detailed in [45], where gamification is analyzed when teaching industrial cybersecurity through Kaspersky Industrial Protection Simulation (KIPS) [46].

In addition to the courses described in recent academic literature, there is hands-on professional training provided by private education organizations that offer different certifications [47]. For example, SANS offers a specialized course on ICS active defense and incident response that leverages tools like Shodan [48].

Online courses (e.g., Massive Open Online Courses (MOOCs)) like the Master Certificate In Cyber Security [49] and novel digital tools are also easily accessible and scalable teaching approaches. Such courses and tools provide a fast and cost-efficient way to deliver location-independent content and even to replace on-site lectures of instructors [29]. However, they do not adjust adequately to the learning capabilities or disciplines of the students (e.g., IT, criminology, legal studies), they do not support team building, communication and interpersonal skills, and they do not reinforce the value of tutoring.

Although there is an increasing number of BSc, MSc, postgraduate and PhD degree programs [50–52], MOOCs, professional and life-long training and digital content [53] in the topics of Industry 4.0 and IIoT cybersecurity, most current programs only provide publicly available generic information (e.g., prerequisites, co-requisites, learning outcomes, syllabus, evaluation criteria, faculty). In addition, there is a lack of literature on teaching/learning effectiveness meta-studies, with only a limited number of papers that focus on specific cybersecurity learning outcome measurements [54]. Since learning outcome evaluation is complex and multidimensional and it involves assessing improvements in performance and an adequate level of engagement and satisfaction, evidence of the different learning outcomes of different courses are hard to measure and compare. Ideally, an evaluation framework for Industry 4.0/IIoT cybersecurity teaching should have a broad scope, be comparative, standardized, specific, flexible, use a multi-level (e.g., individual, team) and mixed approach with qualitative and quantitative measures. Table 1 indicates some key metrics that have been considered during the course for the evaluation at the beginning, during and at the end of the course. Such metrics will ease further comparison with other cybersecurity courses.

**Table 1.** Main characteristics of the outcome measurement proposed.

|  | What? (Qualitative) | What? (Quantitative) | How? |
|---|---|---|---|
| Observed | Behavioral performance, conflicts | Participation, teamwork, soft skills | Labs, tutorials, questionnaires |
| Self-Reported | Context, motivation, engagement | Learning satisfaction | Feedback survey |
| Tested | Technical skills, knowledge | Performance | Final project, final exam, lab reports |

Considering the previously mentioned aspects and works, this article suggests using a blended teaching methodology that proposes:

- To base the practical part on an online tool like Shodan, whose basic functionality can be used by any student with only an Internet connection and a web browser. In addition, no actual class attendance is necessary to access specialized hardware, so in situations like the recent COVID-19 pandemic, students can carry on working remotely.

- To challenge the students and to foster their curiosity by providing real use cases that they have to analyze.
- To present numerous real examples for industrial scenarios in order to raise the student awareness on the severe security problems that anyone with a minimum knowledge can find fast by using tools like Shodan.

## 2.2. IIoT and Industry 4.0 Cybersecurity Tools

A typical cybersecurity audit/attack is composed by four phases:

- Recon. In this first phase information on the target is collected. Such data can be obtained from diverse information sources (e.g., web sites, manufacturers, user manuals, datasheets) and usually involves scanning the target ports to look for potential open services, misconfigurations and vulnerabilities.
- Audit/Attack planning. After analyzing the potential entry points of the target, the auditor/attacker needs to create an audit/attack plan. For such a purpose it is necessary to choose the right steps and tools. Regarding the latter, it is often necessary to develop software tools adapted to the audit/attack scenario so as to exploit the vulnerabilities detected in the previous phase.
- Access. The designed plan is executed by making use of the selected tools in order to access the target system by exploiting the detected vulnerabilities.
- Execution. After gaining access, the designed audit/attack plan will continue to take control of the system. In this phase it is usual to perform different actions to maintain the gained access for future intrusions (e.g., by creating a back door).

Among the four previously described phases, the Recon phase is arguably the most tedious, since it involves dedicating a significant amount of time and resources. Luckily, there are tools that allow for shortening such a phase drastically. Some of the most relevant are compared in Table 2 in terms of their execution requirements, their scanning exposure (i.e., their ability for hiding the auditor/attacker IP/port when scanning a target), their ease of use, their pricing model and the APIs they provide. As it can be observed in Table 2, there are tools that require, besides an Internet connection, only a web browser, which is ideal for setting up cybersecurity labs fast and thus decreasing potential installation problems. Moreover, to prevent potential legal and network problems, students should not carry out indiscriminate scans on industrial devices over the Internet, so the port/vulnerability scanners provided by ZMap [55], Metasploit [56], Nmap [57] or Nessus [58] should be avoided. Among the rest of the compared tools, for the learning approach proposed in this article, Shodan was selected due to its low execution requirements (it only requires an Internet connection and a web browser, so students can use it remotely and even with a smartphone or a tablet), its ease of use (its basic interactions are similar to the ones required when using a web search engine), its available APIs (two developer APIs and an Exploit API are available) and its pricing policy for students and academia, which allows for using most Shodan features for free. Nonetheless, it is important to note that the methodology and use cases proposed in this article can be easily adapted to be used with other tools like Censys [59], ZoomEye [60], BinaryEdge [61] or Onyphe [62].

Shodan has been previously used to teach general IoT cybersecurity through practical use cases, but following a different teaching methodology and without considering the security implications of industrial scenarios [63]. Moreover, Shodan has also been used to assess the cybersecurity of devices that are part of the IT systems of an industrial company. For example, vulnerable webcams, routing devices and firewalls were analyzed by Albataineh et al. [64] after detecting them with Shodan. Web cameras were also studied by Bugeja et al. [65], who found that a significant number of them were weakly protected or not protected at all, which may allow attackers to use them for cyberattacks [66].

Finally, it is worth mentioning that there is a specific area on Shodan's webpage for detecting multiple ICSs [67]. In fact, different researchers have previously warned on the existence of numerous ICSs, Supervisory Control and Data Acquisition (SCADA) systems and PLCs exposed on the Internet and that can be easily discovered with Shodan [68–72]. In addition, Shodan has been used for detecting

CPSs [73] or smart healthcare devices [74], thus demonstrating that little effort is required to detect critical IIoT devices and exposed Industry 4.0 services.

**Table 2.** Comparison of some of the most relevant tools for shortening the Recon phase.

| Tool | Audit/Attack Phase | Basic Execution Requirements | Scanning Exposure | Ease of Use | Dev. API | Exploit API | Pricing Model |
|---|---|---|---|---|---|---|---|
| Shodan | Recon | Web browser | No | High | Yes | Yes | Subscriptions from $59 per month. Free for limited results and reduced functionality. Most advanced features are free for academics. |
| Censys | Recon | Web browser | No | High | Yes | No | Subscriptions for a large number of queries from $99 per month. Free for a small number of queries per month. |
| ZoomEye | Recon | Web browser | No | High | Yes | No | Subscriptions for a large number of results from $70 per month. Free for a limited number of results. |
| BinaryEdge | Recon | Web browser | No | High | Yes | No | Subscription price depends on the number of queries per month (from $10/month for 5000 queries). Free for a limited number of results. |
| Onyphe | Recon | Web browser | No | High | Yes | No | Perpetual subscription for individuals for €59. Free for a limited number of queries per month. |
| ZMap | Recon | Linux, MacOS, BSD | Yes | Medium | No | No | Free (Apache License Version 2.0). |
| Metasploit | Recon, Access, Execution | Linux, MacOS, Windows | Yes | Medium | No | No (but exploits can be added) | Free (open-source) and commercial versions are available. |
| Nmap | Recon | Linux, MacOS, Windows, BSD | Yes | Medium | Yes | No | Free (open-source). |
| Nessus | Recon | Linux, MacOS, Windows | Yes | Medium | Yes | No (but the Dev. API indicates vulnerabilities) | Free trial and commercial version (more than $2000 per year) are available. |

## 3. Blended Teaching Methodology

This article suggests structuring the content of the proposed IIoT cybersecurity course in three theoretical parts:

- Basic industrial cybersecurity concepts and cyberattacks. This first part deals with the main concepts on critical infrastructures, essential services, industrial security policies and cyberattack impact. This part can be illustrated with practical examples of industrial cyberattacks, like the ones performed by Agent.btz, Stuxnet or Night Dragon [75].
- Introduction to ICSs. In this second part the essential concepts on ICSs, PLCs, SCADAs, CPSs and Distributed Control Systems (DCSs) are imparted.
- IIoT and Industry 4.0 cybersecurity. The last part of the course introduces the IIoT paradigm, the different Industry 4.0 technologies and analyzes the most relevant attacks on them.

Although the previously mentioned theoretical content is really specific, students with a minimum knowledge on computers and computer networks usually do not have problems to understand it. However, the proposed methodology obtains its better results with students that come from IT fields,

like computer scientists and electrical engineers, who often have previously learned the essential concepts on computer network architectures and computer security.

The theoretical content is taught following the flipped classroom principles [44], so students receive theoretical content and additional material before the class (e.g., presentations on industrial cybersecurity from the most relevant security conferences, like DEF CON [76], BlackHat [77], or CCC [78]), which are discussed during the face-to-face time.

In addition to the theoretical part, six labs are imparted. In such labs the students first learn how to use Shodan (Lab 1) and then put in practice the learned theoretical concepts on industrial cybersecurity. Thus, the course content is structured according to the following syllabus:

- Week 1: Industrial cybersecurity basics.

  - Learning goals:

    * To learn the essential concepts behind industrial network security.

  - Developed competences:

    * To gain knowledge about legal and technical standards used in industrial cybersecurity, their implications in systems design, in the use of security tools and in the protection of information in industrial scenarios.
    * To gain knowledge about the role of cybersecurity in the design of new industrial processes, as well as of the singularities and restrictions to be addressed in order to build a secure industrial infrastructure.
    * Ability to identify and diagnose the associated risks of cyberattacks on industry and critical infrastructures.

  - Addressed topics:

    * Introduction to industrial cybersecurity.
    * Industrial cybersecurity policies.
    * Impact of cyberattacks on industry and critical infrastructures.
    * Practical industrial cyberattack use cases.
    * Lab 1: Shodan basics.

- Weeks 2 and 3: ICS cybersecurity.

  - Learning goals:

    * To learn the essential concepts on the different types of ICSs and their security.

  - Developed competences:

    * To gain knowledge about how the most popular ICS hardware and protocols work and their role in industrial processes.
    * To gain knowledge about ICS cybersecurity, including the most relevant attacks and potential defense strategies.

  - Addressed topics:

    * Types of ICSs.
    * Traditional industrial communications architectures.
    * Advanced communications architectures.
    * CPSs.
    * Labs 2 and 3: Practical use cases.

- Week 4: IIoT and Industry 4.0 cybersecurity.

– Learning goals:

  * To learn the essential concepts on the security of IIoT and the latest Industry 4.0 technologies.
  * To understand the security implications of making use of Industry 4.0 technologies.

– Developed competences:

  * To gain knowledge about IIoT hardware, software and infrastructure, as well as about their role in industrial scenarios.
  * To gain knowledge about Industry 4.0 technologies and their cybersecurity.

– Addressed topics:

  * Introduction to Industry 4.0.
  * Introduction to IoT and IIoT systems.
  * Main IIoT cyberattacks.
  * Main Industry 4.0 cyberattacks.
  * Lab 4: Advanced Shodan scripting.

- Weeks 5 and 6: Practical Industry 4.0 cyberattacks.

  – Learning goals:

    * To understand the main network security issues, and the different protection techniques and attacks for Industry 4.0 systems, as well as to know how to implement them.

  – Developed competences:

    * To gain knowledge of cyberattack and cyberdefense techniques on Industry 4.0 systems.
    * To gain the ability to apply theoretical knowledge to practical situations within the scope of industrial infrastructures, equipment or specific application domains, including precise operating requirements.
    * Ability to innovate and contribute to the advance of industrial cybersecurity by designing new algorithms or techniques for industrial devices in order to eventually help in the protection public, private or commercial of industrial assets.

  – Addressed topics:

    * Cyberattacks to industrial robotics, Unmanned Aerial Vehicles (UAVs), Automatic Guided Vehicles (AGVs) and Autonomous Underwater Vehicles (AUVs).
    * Augmented/Mixed/Virtual Reality cybersecurity.
    * Cloud/edge/mist computing cybersecurity.
    * Blockchain and Distributed Ledger Technology (DLT) cyberattacks.
    * Auto-identification system cybersecurity.
    * Vertical and horizontal integration system cybersecurity.
    * Additive manufacturing cyberattacks.
    * Labs 5 and 6: Final project.

As it can be observed in the previous syllabus, the content is imparted during an intensive six-week course, which corresponds to a 3 ECTS (European Credit Transfer System) credit workload that belongs to the Master Program on Cybersecurity delivered jointly by the universities of A Coruña and Vigo [79]. The course is attended by university students with diverse backgrounds, but mostly by computer scientists and electrical engineers.

Each week of the course, three hours are dedicated to lectures and practical labs. Moreover, the students had to do a final project on IIoT/Industry 4.0 security whose main topic was selected

by them (for instance, some students of the 2020 class analyzed adversarial attacks [80] for industrial robotics or smart grid meter security). Thus, during the course the students deliver two lab reports and a final project memory together with all the involved software. Overall, the course was designed for requiring 20 hours of in-person activities and 54 hours of autonomous work.

The following are the main learning outcomes expected from the course:

- To learn the essential concepts on industrial cybersecurity.
- To understand the different defense and attack techniques that affect industrial systems and to know how to implement them.
- To understand the most common security problems and attacks that affect industrial networks, as well as to know the essential techniques to minimize them.
- To be able to understand the impact and security implications involved in the deployment of the latest Industry 4.0 technologies.

The learning outcomes related to learning concepts are evaluated through a theoretical final exam (40% of the grade), while the rest of the outcomes are assessed through the lab reports (30%) and the final project (30%).

## 4. Shodan for Practical IIoT/Industry 4.0 Cybersecurity Labs

### 4.1. Lab 1: Shodan Basics

Shodan is an Internet scanner that searches for text banners, which are shown by a device when a remote user connects to them (for instance, the welcome message of an FTP server) [81]. Such banners usually contain information that allow for identifying certain characteristics of an Internet service (e.g., the software version of an FTP server) or of the computer where it runs (e.g., the version of the operating system). The collected information can be used later for detecting vulnerabilities and for defining potential cyberattacks.

Shodan can be used like other web search engines, but the results it produces are conditioned by the type of user that performs the query. For instance, non-registered users have a limited number of queries and cannot use certain Shodan features. One of such features are filters, which allow for narrowing down search results geographically (e.g., by country, city), by IP (e.g., specific IP or network), by port or organization [82].

An example of Shodan search with filters is illustrated in Figure 2, which also indicates the different parts of a Shodan main result page. As it can be observed in Figure 2, the result page lists more than 22,000 devices in which port 102 is open. Such a Shodan query is really relevant for the industrial field, since it allows for discovering Siemens S7 PLCs (port 102 is used by the S7 protocol to communicate PLCs of the Siemens S7 family).

It is important to indicate the students that not all the discovered devices are 'real' S7 PLCs: there is a significant amount of honeypots that are deployed with the goal of determining when cyberattacks are performed, thus gathering information on the strategies of the attackers. This is a common problem in previous Shodan literature: statistics on detected devices are given based on a specific query, but honeypots are not filtered out. For instance, the previous Siemens S7 query can be refined to determine which of the previous detected devices were actually making use of the ICS/SCADA honeypot Conpot [83]. It is also worth mentioning that Shodan provides a tool that helps to identify to a certain extent (i.e., with an estimated probability called 'Honeyscore') whether an IP address is a honeypot [84].

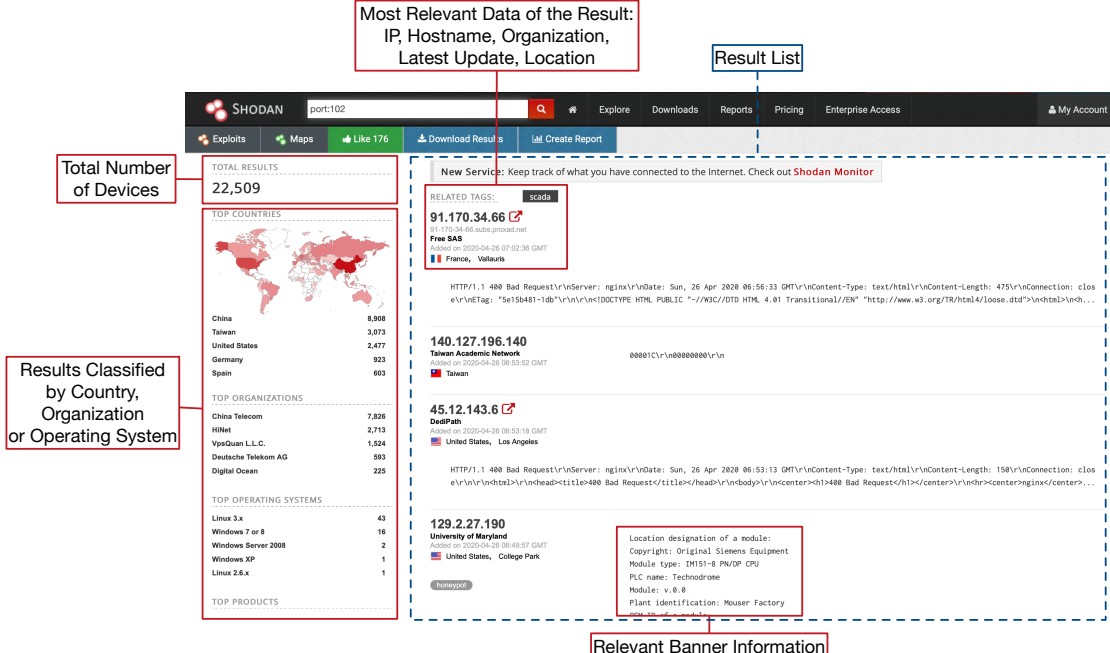

**Figure 2.** Shodan result list of potential Siemens S7 PLCs.

*4.2. Labs 2 and 3: Practical Use Cases*

After the students complete the lab on the basics of Shodan, they carry out the second lab of the course. Depending on the previous knowledge and skills of the students, such a second lab can follow two different approaches:

- Guided approach:

    1. First, the course instructor gives each student a list of Shodan queries like the ones detailed in Appendix A.
    2. The students then try the queries on Shodan and analyze the obtained results in order to determine which IIoT or Industry 4.0 devices are detected by the queries. Although in many cases the information gathered by Shodan is enough to identify the devices, the students usually need to make use of additional web search engines to find information from the manufacturers or distributors.
    3. Once the students gather enough information to identify the IIoT/Industry 4.0 devices and understand their essential inner working, they analyze the vulnerabilities and exploits detected by Shodan. Moreover, the students look for additional signs of weak security, like the use of default credentials or outdated security mechanisms.

- Autonomous approach:

    1. In this approach the course instructor initially gives the students a specific target or a set of targets (i.e., IIoT/Industry 4.0 devices).
    2. The students first investigate the targets by gathering information from their datasheets or manuals with the objective of understanding how they work and which are their default credentials.
    3. Then the students build Shodan queries that allow for discovering the targeted devices.
    4. Finally, the vulnerability assessment is performed with the help of the Shodan exploit API and other data sources like the available Common Vulnerability and Exposure (CVE) reports.

As an example, Table 3 provides a summary of some of the results obtained by the students for different use cases analyzed during the 2020 course. The main conclusions for such use cases are:

**Table 3.** Examples of the results obtained by the students.

| | C4MAX Vehicular Monitor | NetBotz Monitor | Somfy Alarm System | Proliphix Thermostats | Automatic Tank Gauges | Electro Industries Gaugetech Smart Grid Meter |
|---|---|---|---|---|---|---|
| #Shodan Results | 654 | 103 | 17,294 | 192 | 1621 | 59 |
| #Analyzed Devices | 20 | 20 | 20 | 20 | 20 | 20 |
| #Devices without Authentication | 20 | 7 | - | - | 20 | 14 |
| #Devices with Default Credentials | - | - | 2 | 3 | - | 2 |
| #Devices Affected by CVEs | 1 | 2 | - | - | - | - |
| #Detected CVEs | 9 | 2 | - | - | - | - |

- C4MAX IIoT monitoring devices:

    - Shodan query: *"[1m[35mWelcome on console"*.
    - All the devices were completely open through their Telnet service.

- NetBotz monitor:

    - Shodan query: *"Netbotz appliance"*.
    - Roughly 40% of the detected devices have no security at all and allow for accessing the environmental data and onboard web cameras. The rest of the devices required authentication.

- Somfy alarm system:

    - Shodan query: *title:"Centrale" "Pragma: no-cache, no-store"*.
    - Most alarm systems are properly configured, a couple of them made use of the default administration credentials, so any user with access was able to switch off or on the alarms.

- Proliphix thermostats:

    - Shodan query: *title:"Status & Control"*.
    - All the analyzed devices control panels were open and provided the ambient temperature without requiring credentials.

- Automatic tank gauges:

    - Shodan query: *unleaded*.
    - All the analyzed devices provided the tank gauge data without carrying out an authentication procedure. In fact, such data could be obtained through Shodan (i.e., no direct connection with the devices was required to gather the IIoT data).

- Electro Industries Gaugetech smart grid meter:

    - Shodan query: *"Server: EIG Embedded Web Server" "200 Document follows"*.
    - Most devices provided barely any security in spite of monitoring a critical infrastructure like the electrical grid. Thus, 80% of the analyzed smart meters could be accessed as administrator.

### 4.3. Lab 4: Advanced Shodan scripting

Although Shodan provides a really clean and intuitive web interface for finding IIoT/Industry 4.0 devices fast, in practice, vulnerability assessment automation requires making use of Shodan APIs. There are Shodan APIs for multiple languages (e.g., Python, Java, Ruby) and are offered either as Representational State Transfer (REST) APIs or streaming APIs. The former enables exchanging information with Shodan through GET, POST, DELETE and PUT requests, while the latter uses JavaScript Object Notation (JSON) files for the data exchanges. In addition, Shodan provides an exploit REST API that collects vulnerabilities from diverse data sources.

As an example, Listing 1 shows a Python script that automates the detection of potential Siemens S7 PLCs and their exploits. The script, which is also provided as supplementary material, first prints the IP and city of the devices and then gets the detected vulnerabilities and exploits through Shodan exploit REST API. It is important to note that, to make use of the script, the developer needs to indicate a valid Shodan API key. Moreover, the *sleep(1)* calls should not be removed, since a one-second delay is necessary to respect the Shodan query limit.

Listing 1: Python script for automating Siemens S7 PLC exploit detection.

```python
import shodan
from time import~sleep

SHODAN_API_KEY = ''[PUT KEY HERE]''
api = shodan.Shodan(SHODAN_API_KEY)
query = 'port:102'

try:

    PLCs = api.search(query)
    print('Total number of Siemens S7 PLCs: {}'.format(PLCs['total']))

    for result in PLCs['matches']:

        print('IP: {}'.format(result['ip_str']))
        host = api.host(result['ip_str'])
        print('- City: {0}'.format(host.get('city', 'n/a')))
        print('')
        sleep(1)
        try:
            if str(host.get('vulns')) != 'None':
                print('-------------- Detected Exploits --------------------')
                for vulnerability in host.get('vulns'):
                    exploits = api.exploits.search(vulnerability)
                    sleep(1)
                    print('Found {0} exploits for vulnerability ''{1}'' \n'.format(
                    exploits.get('total'), vulnerability))

        except shodan.APIError as erro:
            print(
            'Error found for exploit query: ''{0}'''.format(query))
            print('Shodan error: {0}'.format(erro))

except shodan.APIError as e:
    print('Error: {}'.format(e))
```

## 5. Teaching Results

The methodology and contents described in this article have been used by the authors during the last years for teaching an IIoT and Industry 4.0 course to 119 students. Due to the background of the students (they had no previous experience on industrial cybersecurity), of the two possible versions of the methodology described in Section 4.2, the guided approach was selected. Thus, every student first

had to investigate three different Shodan queries assigned by the instructor and then automate the search through Python scripts like the one previously detailed in Section 4.3.

It is important to note that all the students that attended the four courses were explicitly asked to not to exploit the detected vulnerabilities or to carry out brute-force attacks on the devices (i.e., the students' goal was to audit the devices to learn how to apply cybersecurity by design), which would probably raise the number of detected compromised devices significantly. Since it is difficult to control the behavior of all the students (who may be tempted to exploit certain vulnerabilities), to prevent potential legal problems and dangerous situations, an instructor should select the course practical use cases carefully, thus avoiding critical infrastructure and other industrial systems in which the modification may derive serious consequences.

In addition, universities should check the legal requirements of putting in practice a course like the one proposed, since law differs significantly from one country to another. For instance, in the country where the proposed course was imparted (Spain), a mere vulnerability scan over a critical infrastructure usually derives into a complaint from the entity that manages the infrastructure. Such a complaint is generally settled in a friendly way, since no damage is caused. However, if damage exists, the action may be classified as a cyberattack and thus derive into a formal police report. Therefore, it is extremely important to warn the students on what they are going to find and on the impact that their actions may have from the physical damage and legal consequences point of view. As a consequence, before giving the students any industrial target to be analyzed, it is essential to establish a warning mechanism that involves the instructor: the students should never interact with the discovered industrial devices and, if they find a relevant exposed industrial system, they should warn the course instructor, who will take the necessary measures.

## 5.1. Obtained Results

This section summarizes the results obtained by the 119 students that took the last four editions of the proposed course. A total of 119 students took part in the course (105 men and 14 women), which were mostly computer scientists, electrical engineers and electronic engineers. The vast majority were advanced programmers, but they had neither previous knowledge of Shodan or practical experience on IIoT or in most Industry 4.0 technologies.

The following are the general results obtained during the four imparted courses:

- A total of 99 students eventually took the whole course (20 students dropped out during the course), delivering 198 reports for the two practical labs.
- A total of 178 different queries were given to the students, of which 118 were eventually analyzed (60 of them were assigned to the 20 drop-outs, but they are not considered in the presented results) and actually targeted 47 different IIoT and Industry 4.0 devices (some of the queries targeted the same devices).
- For each of the 118 Shodan queries, only the first 20 Shodan results were analyzed. This means that a total of roughly 2360 devices were studied.
- For the 47 studied IIoT/Industry 4.0 devices, a total of 3749 vulnerabilities related to already published CVEs were found by the students. Such a high number was mainly due to a few IIoT/Industry 4.0 devices that executed outdated software (e.g., outdated PHP versions, old HTTP servers).
- Of the 2360 studied devices, 205 implemented no authentication mechanisms, while 103 of them made use of the default administration credentials. These results imply that cyberattackers may easily take control over 13% of the analyzed industrial devices. It is relevant to point out that, if the results obtained by the 2020 class were only considered, the mentioned percentage would be roughly 25%, since 220 out of the 900 industrial devices analyzed by the students did not make use of proper security mechanisms.

As a conclusion of the imparted courses, it can be stated that, despite the students' lack of previous knowledge on industrial cybersecurity, it was possible to train them fast to detect potential

vulnerabilities on IIoT and Industry 4.0 devices connected to Internet. In addition, the students concluded that Shodan is a simple but really powerful tool for cybersecurity researchers that emphasizes the need for taking care of the security of the exposed devices, especially for industries that are considered critical.

With respect to the problems that the students found during the courses, they were mainly related to two aspects:

- Shodan API programming. Some of the students had problems during the development of the Python script. Specifically, the following were their most frequent problems:

  - Excessive number of requests. For most account types, Shodan restricts the number of requests to one per second, so it is necessary to include a software delay in the script (in other case Shodan will return an error).
  - Use of filters in scripts. Some students had problems when making use of certain filters through Shodan's programming API because the account type they were using did not allow such a use. There are two potential solutions to this problem: to upgrade the account type or to perform the search in two sequential steps: the first step would gather the data on a specific query, while a second one would parse the collected data according to a certain filter. Obviously, this latter method is clearly slower than the former, since it requires to parse the collected results locally with the developed script.
  - Lack of documentation. Shodan's REST API documentation is clear, but the documentation on some of the wrappers that allow for using it with certain programming languages is not so complete. In the case of the official Python wrapper [85], although it has improved over the last years and provides good examples on basic searches, some content is still missing, like the meaning of certain fields that enable accessing relevant data. For instance, students had problems for determining which information was available on the detected hosts and how to access it (e.g., it is not straightforward to determine that the vulnerability information on a host is accessed through the 'vuln' field).

- Lack of knowledge on the analyzed industrial equipment. The students had to look for manufacturer manuals and datasheets in order to understand how the studied devices operated and then determined the impact of potential vulnerabilities.

*5.2. Overall Marks and Learning Outcomes*

As it was previously indicated in Section 3, the students are evaluated through a theoretical exam, lab reports and a final project. The final grade is given within the range 0 to 10, which is the most commonly used in the Spanish university system.

Table 4 shows the average, median and variance of the grades of all the students that took part in the four editions of the course. The overall average course grade was 8.1813. As it can be observed in Table 4, the grades are in general high, but lower for the final exam. These results reflect an aspect that was also observed by the course instructors: students are more motivated by the practical part of the course than for the theoretical concepts behind it.

In addition, it is interesting to analyze the correlation between the different grades. The covariance analysis indicates that there is a positive correlation between the grades of the lab reports and the one obtained for the final project (0.401) and between the final project and the exam (0.226), but there is no correlation between the grades of the lab reports and the final exam grade ($-0.067$). Therefore, the results indicate that the students that delivered good lab reports also delivered a good final project and those performed well in the final exam. However, although the course practical and theoretical parts are related, they evaluate different industrial cybersecurity aspects, so good practical skills do not guarantee to obtain a high grade on the final exam. In any case, at the view of the results, it can be concluded that the main learning outcomes are accomplished:

- The students demonstrate at the end of the course that they know the essential concepts on industrial cybersecurity.
- The students are able to apply successfully the course methodology to evaluate through Shodan the security of real industrial systems.
- The students also demonstrate through the lab reports and final projects that they understand the theory behind it and thus they are able to address the most common cybersecurity problems and attacks that affect IIoT/Industry 4.0 systems.
- The students end up being aware of the dangers of exposing insecure industrial devices on the Internet and of the impact derived from the use of non-properly protected devices of diverse Industry 4.0 technologies.
- Due to time and workload constraints, part of the work carried out by the students in their supervised projects was restricted to specific regions (e.g., countries, cities, industrial regions) or industries so as to limit the diversity of analyzed devices and thus get in-depth knowledge on them. Such analyses can provide high-value advice to different industrial companies and sectors. In addition, a cross-comparison of the practicality for different industries and countries can also provide some interesting insights, especially from a business standpoint.

It would have been very interesting to perform an additional control questionnaire about the learning outcomes a few months after finishing the course but, unfortunately, it could not be carried out due to legal restrictions. Specifically, once a course is finished, according to the Spanish law of Protection of Personal Data (Ley Orgánica de Protección de Datos de Carácter Personal, LOPD) and the Regulation EU 2016/679 [86] followed by the University of A Coruña [87], student contact information should be removed.

**Table 4.** Average, median and variance of the grades obtained by all the students that took the course.

|          | Lab Reports | Final Project | Final Exam |
|----------|-------------|---------------|------------|
| Average  | 8.651       | 8.884         | 7.302      |
| Median   | 8.625       | 9.000         | 7.437      |
| Variance | 0.802       | 1.414         | 2.705      |

*5.3. Student Feedback*

Regarding the feedback from the students, it was collected by the university quality measurement system, which assigns a grade between 1 and 7 to different aspects of the course: '1' means that the students completely disagree with the statement, while a '7' means that they completely agree. It is important to note that the university quality measurement system was not available for the first edition of the course. For the rest of the course editions, only 22% of the 119 students answered the feedback survey. Such a survey was collected directly through the university quality measurement system ("Avaliación docente"), which acts as an independent evaluator of each of the courses imparted at the University of A Coruña. In addition, as a second step, the responses and the instructor are evaluated by third-party agencies that ensure education quality, like the Spanish University System Quality Observatory of the National Agency for Quality Assessment and Accreditation (ANECA) and the Agency for Quality Assurance of the Galician University System (ACSUG). It must be noted that the achieved percentage of response, although it may seem low, it is actually significantly higher than the average student response percentage for the Master program where the course is imparted, which in the last years has only received responses from 13.04% of the students [88]. This is essentially due to the fact that filling the questionnaires is a voluntary activity.

Specifically, the following questions were asked:

1. The course was properly structured.
2. The amount of work was proportional to the assigned hours.
3. The course required guidance from the instructor.
4. It was necessary to ask the instructor doubts on the course lectures or labs.

5. The evaluation methodology was appropriate considering the imparted content.
6. With the content imparted in this course I reached my own learning objectives.
7. I am globally satisfied with the course.

The obtained results are shown in Table 5 and the distribution of the answers for each value of the Likert scale is shown in Table 6. Additionally, a bar chart of the results can be seen in Figure 3. At the view of the feedback results, it can be concluded that the students perceived the course as well structured and organized (questions 1 and 2), appropriate for carrying out part of the work autonomously (questions 3 and 4), properly evaluated (question 5) and satisfactory in general and in terms of their learning goals (questions 6 and 7).

**Table 5.** Results obtained through the course feedback survey.

|  | **Question 1** | **Question 2** | **Question 3** | **Question 4** | **Question 5** | **Question 6** | **Question 7** |
|---|---|---|---|---|---|---|---|
| Average | 6.46 | 5.85 | 4.54 | 5.38 | 6.92 | 6.58 | 6.46 |
| Mode | 7 | 7 | 4 | 7 | 7 | 7 | 7 |
| Median | 7 | 6 | 4 | 6 | 7 | 7 | 7 |

**Table 6.** Number of answers obtained for each value of the Likert scale.

|  | **1** | **2** | **3** | **4** | **5** | **6** | **7** |
|---|---|---|---|---|---|---|---|
| Question 1 | 0 | 0 | 0 | 0 | 2 | 10 | 14 |
| Question 2 | 0 | 0 | 2 | 0 | 8 | 6 | 10 |
| Question 3 | 0 | 2 | 4 | 8 | 4 | 6 | 2 |
| Question 4 | 0 | 0 | 6 | 2 | 4 | 4 | 10 |
| Question 5 | 0 | 0 | 0 | 0 | 0 | 2 | 24 |
| Question 6 | 0 | 0 | 0 | 2 | 0 | 5 | 19 |
| Question 7 | 0 | 0 | 0 | 0 | 4 | 6 | 16 |

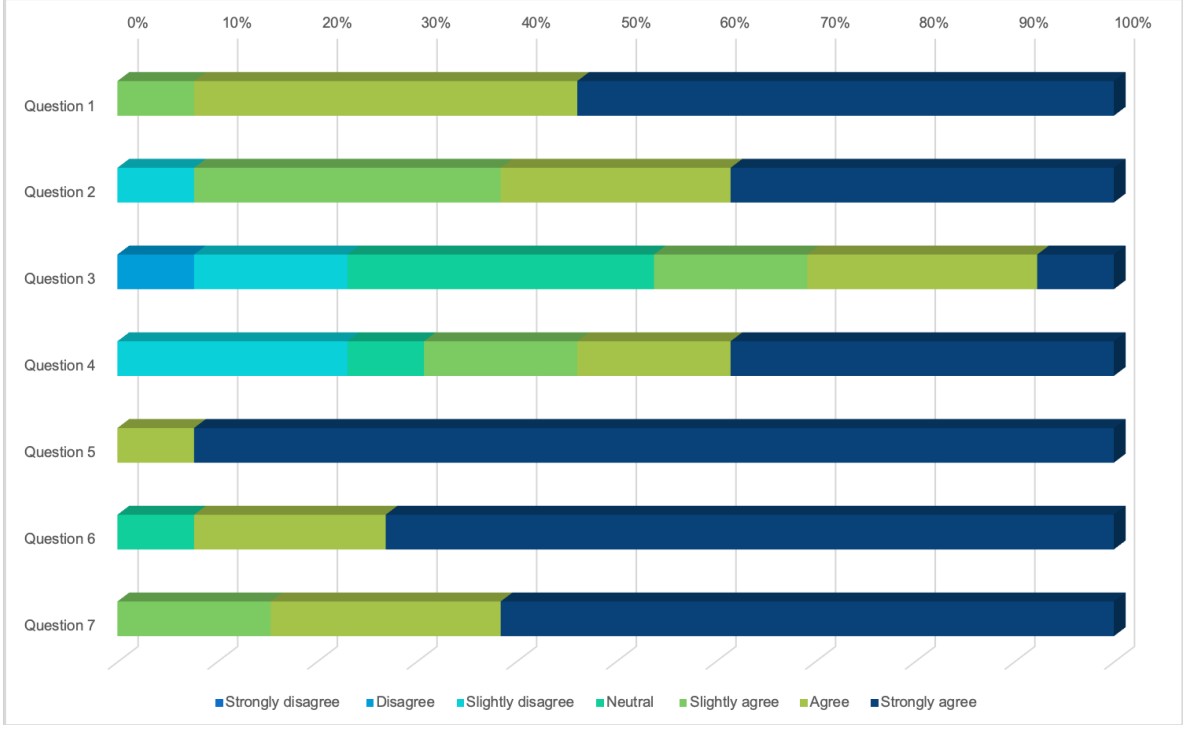

**Figure 3.** Bar chart of the feedback survey results.

## 6. Conclusions

The interest in IIoT and Industry 4.0 cybersecurity has increased in the last years, but, unfortunately, such fields are not currently widely taught at a university level. To tackle such an issue, this article

proposed a practical use case-based methodology that enables performing industrial cybersecurity audits fast on devices exposed on the Internet. The described blended teaching methodology is aimed at IT university students but requires no previous experience on industrial cybersecurity tools thanks to tools like Shodan, which provide a search engine able to find IIoT/Industry 4.0 devices in an intuitive and easy way. Thus, this article detailed multiple online searches that allow for discovering thousands of devices related to the different Industry 4.0 technologies. The results obtained during the four editions of the course show the usefulness of the proposed methodology: the students were able to find that 13% of the analyzed IIoT/Industry 4.0 systems could be easily accessed and that such industrial systems had 3749 CVE-related vulnerabilities. Moreover, the feedback received from the students on the course was really positive. In addition, the blended methodology was flexible, even during the COVID-19 lockdown, and the obtained teaching results indicate that the established course learning outcomes were accomplished and, as a consequence, the targeted competences can be considered to have been acquired. Therefore, this article provides useful guidelines for teaching industrial cybersecurity through practical use cases, which will allow for training the next generation of security researchers and developers of the Industry 4.0 era.

**Supplementary Materials:** The following code is available online at http://www.mdpi.com/2076-3417/10/16/5607/s1, Listing 1: Python script for automating Siemens S7 PLC exploit detection.

**Author Contributions:** T.M.F.-C. and P.F.-L. contributed equally to the involved analysis and writing. T.M.F.-C. conceived the article and performed the data collection. All authors have read and agreed to the published version of the manuscript.

**Funding:** This work has been funded by the Xunta de Galicia (ED431G 2019/01), the Agencia Estatal de Investigación of Spain (TEC2016-75067-C4-1-R, RED2018-102668-T, PID2019-104958RB-C42) and ERDF funds of the EU (AEI/FEDER, UE).

**Conflicts of Interest:** The authors declare no conflict of interest.

## Appendix A. Practical IIoT and Industry 4.0 Use Cases for Labs 2 and 3

This Appendix provides numerous examples of practical use cases for Labs 2 and 3 that allow for discovering specific industrial devices that in many cases monitor and control critical infrastructure. It is important to note that, to provide easy-to-test examples, the following subsections indicate specific Shodan queries, which can be easily adapted to be used with the online search engines cited in Section 2.2.

### Appendix A.1. IIoT Devices

Shodan is able to find a significant amount of IIoT devices that can become potential targets for cyberattacks or may be used to perform cyberattacks. One of the types of devices that have been traditionally targeted by Shodan researchers are webcams and Digital Video Recorder (DVR) systems, which are used by many companies in their surveillance systems. Unfortunately, as of writing, many webcam and DVR systems remain unprotected or weakly protected, being really easy to find video-surveillance systems that make use of their default credentials or that do not implement any security mechanism. For instance, such weaknesses were exploited in 2016 by Mirai, a botnet that infected hundreds of thousands of IoT devices like DVRs, webcams or routers to perform one of the largest Distributed Denial of Service (DDoS) attacks in history [89]. Specifically, Mirai was able to create its botnet by performing brute force attacks on weakly protected IoT devices, thus obtaining their management credentials.

A practical example of weakly protected video-surveillance systems that can be found with Shodan are thousands of ExacqVision systems (Shodan query: *"server: wfe"*): a relevant amount of such systems make use of the default administration credentials or of easy-to-guess passwords (e.g., '1234', 'admin'). As an example, Figures A1 and A2 show two screenshots of unprotected Exacqvision systems found with Shodan. There are many more examples of weakly protected webcams (either deployed in industrial scenarios or private households), like the ones that result from searching

for Linksys WVC80N cameras (Shodan query: *WVC80N*), AXIS webcams (Shodan query: *"port:80 has_screenshot:true"*) or AVTECH IP webcams (Shodan query: *linux upnp avtech*).

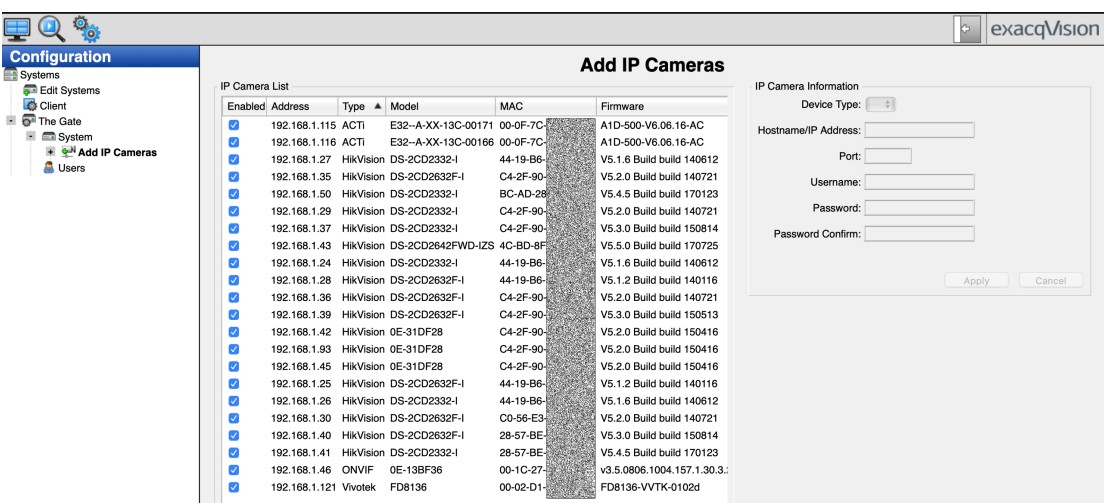

**Figure A1.** Example of a weakly protected Exacqvision configuration video-surveillance system.

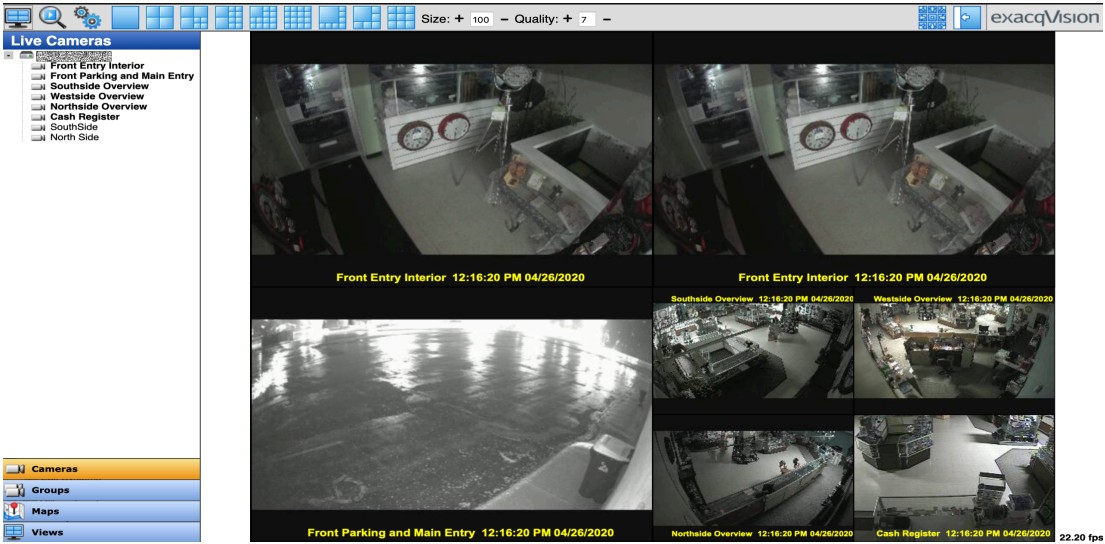

**Figure A2.** Example of another weakly protected Exacqvision video-surveillance system.

Similarly to webcams, unprotected DVR systems can be easily found with Shodan. An example of such systems are UNIMO UDR DVRs (Shodan query: *html:"DVR_H264 ActiveX"*): almost all of the systems analyzed by the authors for this article made use of the default credentials (a screenshot of one of such systems is shown in Figure A3).

Shodan is able to detect other IIoT systems that can be either used by industrial companies or private homes, like certain home automation or climate control systems. For instance, the Somfy system (Shodan query: *title:"Centrale" Pragma:"no-cache, no-store"*), allows for managing a home or company security alarms remotely. Although the Somfy system is password protected through a web interface, unfortunately, it is easy to find with Shodan systems that make use of the default credentials (an example of one of such systems is shown in Figure A4). Regarding climate control system, it is not difficult to find exposed thermostats from ICY (Shodan query: *title:"ICY Clever Thermostat"*), Heatmiser (Shodan query: *title:"Heatmiser Wifi Thermostat"*) or Proliphix (Shodan query: *title:"Status & Control"*). In addition, some control access systems are also exposed on the Internet and part of them use the default administration credentials (e.g., HID VertX systems (Shodan query: *"HID VertX" port:4070*)).

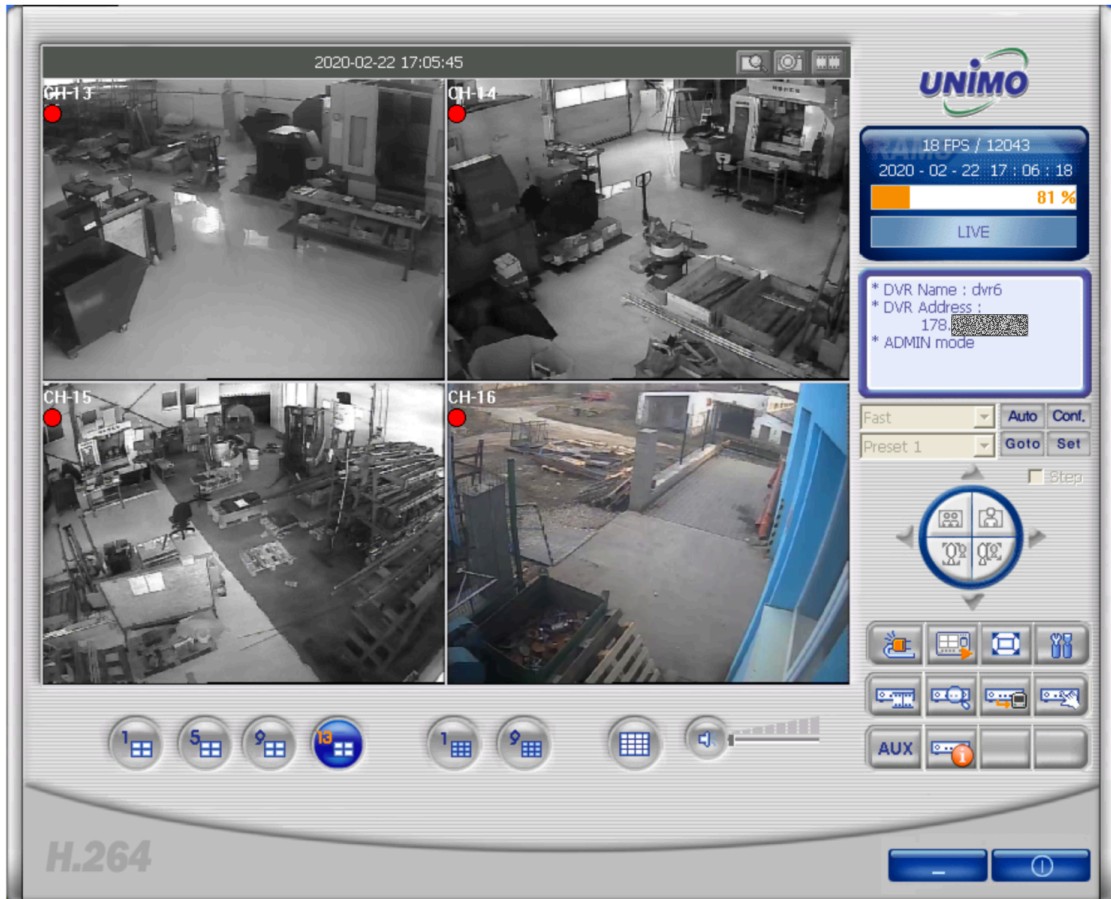

**Figure A3.** Screenshot of vulnerable Unimo DVR system.

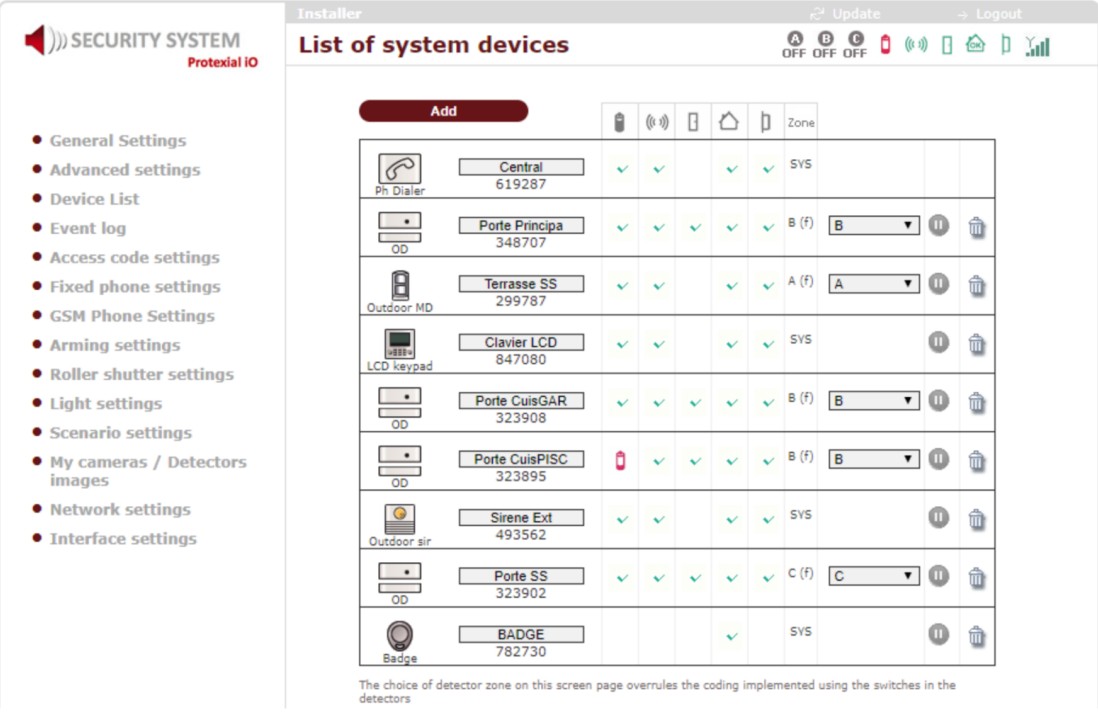

**Figure A4.** Screenshot of vulnerable Somfy alarm system.

It is also worth mentioning that Message-Queue Telemetry Transport (MQTT) protocol is increasingly being used for resource-constrained IIoT devices (e.g., sensors, actuators), but, unfortunately, many deployments lack a proper security configuration. Thus, it is straightforward to discover MQTT brokers (Shodan query: *"MQTT Connection Code: 0" set –alarm*) that are unprotected and that expose the internal MQTT topics.

Industrial vehicles can also be monitored in order to optimize the processes of an Industry 4.0 smart factory. For such a purpose, the GPS and connectivity solution provided by the C4MAX device can be really useful. Unfortunately, most of the C4MAX devices detected by Shodan (query: *"[1m[35mWelcome on console"*) provide a Telnet connection that requires no credentials, which allows remote attackers to access the system and the vehicle data.

Finally, it is worth mentioning that certain voice communications mechanisms that are commonly used by companies can be found by Shodan, like Voice-over-IP (VoIP) Public Branch Exchange (PBX) systems (example of Shodan query: *PBX "gateway console"–password port:23*) or the popular Polycom videoconference systems (Shodan query: *http.title:"–Polycom" "Server: lighttpd"*), in which the management software also suffers from vulnerabilities and the lack of a proper security configuration (e.g., the use of default administration credentials).

*Appendix A.2. Robotics*

Robotic systems are typically used for automating tasks in smart factories, either for operating autonomously (e.g., for painting or soldering tasks) or for collaborating with industrial operators during specific tasks (e.g., to hold heavy objects).

Robotic arms are commonly used in industry and they essentially consist of a controller, different sensors and multiple servo motors that conform the arm. The robotic arm can be controlled manually by an operator or it can act autonomously, according to certain software-defined behavior. In this latter case the controller receives inputs from the sensors (e.g., from a video camera), process them and then decides the actions to be carried out by the servo motors. Cybersecurity problems may arise when the robot controller is connected to the Internet, since a remote attacker may take control of the robot and then cause product or physical damage [90].

Shodan is able to find just a few hundreds of industrial robots connected directly to the Internet, like the ones from ABB Robotics (Shodan query: *ABB slave*), FANUC (Shodan query: *FANUC FTP*), Yaskawa (Shodan query: *Yaskawa*) or Mitsubishi (Shodan query: *Mitsubishi FTP*). However, it must be noted that, in many cases, the robot controller is not connected directly to the Internet, but through a communications router. Shodan is able to find thousands of such routers [90] from manufacturers like Ewon (Shodan query: *ewon*), Westermo (Shodan query: *westermo*) or Sierra Wireless (Shodan query: *sierra wireless*).

*Appendix A.3. ICSs, SCADAs and CPSs*

As it was previously mentioned, Shodan includes a specific section on its web page that provides a good compilation of queries for finding ICSs and SCADAs [67]. For example, it is really easy to find PLCs that make use of Modbus (Shodan query: *port:502*), EtherNet/IP (Shodan query: *port:44818*) or DNP3 (Shodan query: *port:20000 source address*).

ICSs can be monitored or controlled remotely through SCADAs that may not provide an Internet-connected interface. In such cases, many industrial companies allow for accessing the SCADA through a Virtual Network Computing (VNC) system. Shodan enables locating VNC systems really fast (Shodan query: *has_screenshot:true product:VNC "authentication disabled"*). Unfortunately, many developers do not implement any authentication mechanism, like it is shown in Figures A5 and A6 for two SCADAs found with Shodan.

There are also certain cases where a SCADA is initially secured properly, but, due to the lack of software maintenance (i.e., periodic software updates), it becomes vulnerable. A good example are a

significant number of Nordex wind farm management systems (Shodan query: *Jetty 3.1.8 "200 OK"*) that run on an insecure version of Jetty.

Shodan is also able to discover CPSs that are poorly configured in terms of security. For instance, NetBotz is a solution for monitoring remotely the health of data centers. A few NetBotz instances can be detected by Shodan (Shodan query: title:"netbotz appliance") and, as of writing, all are open, so anyone can see the environmental conditions of the data center and, in some cases, the video-surveillance cameras can be accessed (an example is shown in Figure A7 on the left). In addition, certain sensor data that are periodically collected by remote CPSs can be accessed easily through Shodan. For example, queries like *fuel oil*, *diesel* or *unleaded* enable accessing information from automatic tank gauges [91] (an example of the collected data is shown in Figure A7 on the right).

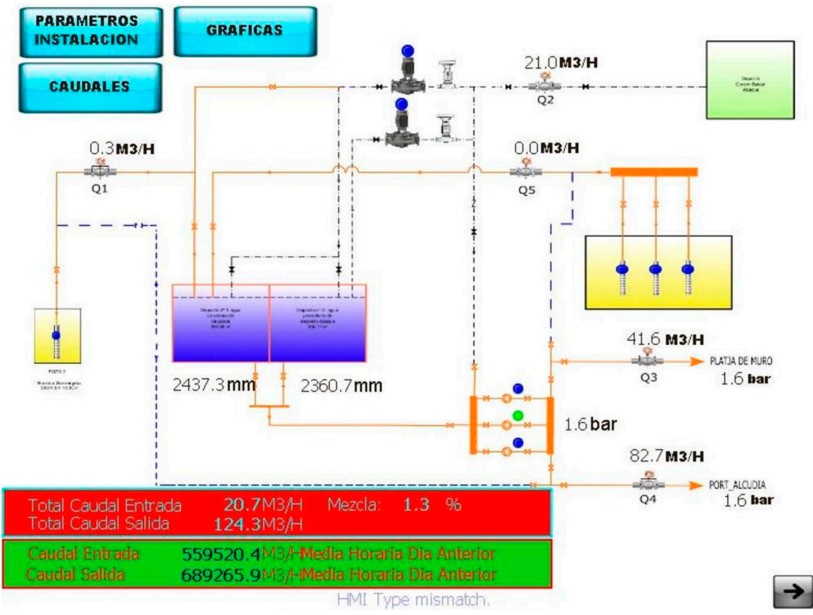

**Figure A5.** Screenshot of open VNC-controlled SCADA found with Shodan.

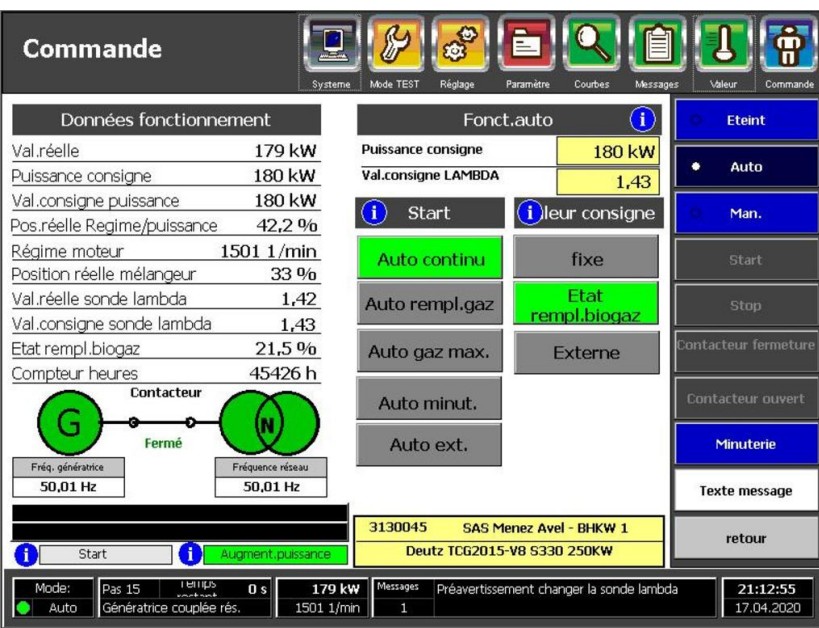

**Figure A6.** Screenshot of open VNC-controlled SCADA found with Shodan.

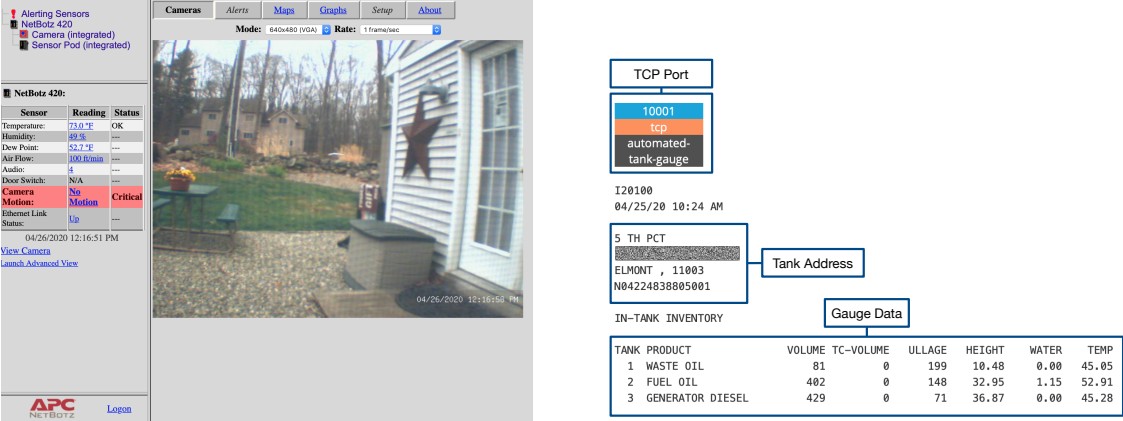

**Figure A7.** Open NetBotz installation (left) and data from an automatic tank gauge (right).

*Appendix A.4. Additive Manufacturing*

Additive manufacturing (also commonly known as 3D printing) systems can also be detected by Shodan, since some of them are exposed on the Internet in order to allow remote users to monitor and control the manufacturing process. For the same reason, other computer-based tools like lathes, milling machines or laser cutters may also be detected by Shodan. For instance, thousands of 3D printers that make use of the open-source software OctoPrint (mostly non-industrial 3D printers) can be found by Shodan (query: *title:octoprint 200 OK*). Most of them are properly secured, but a few hundred of them still remain completely open (Shodan query: *title:"OctoPrint"–title:"Login" http.favicon.hash:1307375944*). Figure A8 shows a screenshot of one of such unprotected 3D printers.

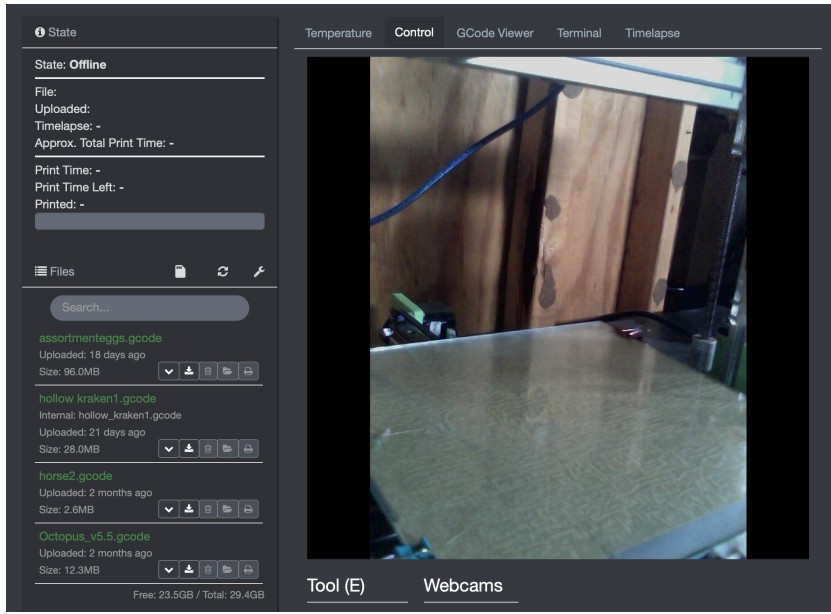

**Figure A8.** Example of exposed 3D printer found with Shodan.

*Appendix A.5. Big Data Software*

Big data software allows for processing and storing large amounts of information that is collected from diverse sources. Once stored, such data are analyzed and then certain conclusion can be drawn and decisions can be taken.

First, it must be noted that Shodan allows for detecting running instances of traditional SQL databases like MySQL (Shodan query: *product:MySQL*) or PostgreSQL (Shodan query: *product:PostgreSQL*). In addition, hundreds of thousands of NoSQL databases can be found executing popular software like

MongoDB (Shodan query: *product:MongoDB*), CouchDB (Shodan query: *product:CouchDB*), Cassandra (Shodan query: *product:Cassandra*) or Elasticsearch (Shodan query: *port:9200 json*). In the case of MongoDB, it is straightforward to find open databases (Shodan query: *product:MongoDB databases*), most of which have been already hacked (this can be observed since there is a database called 'READ_ME_TO_RECOVER_YOUR_DATA' or similar that in its content demands Bitcoins to restore the database data). Regarding Cassandra, Shodan finds a few hundred open databases whose keyspaces are exposed (Shodan query: *product:cassandra keyspaces*).

Probably, the most popular software tool for processing and managing large amounts of data is Apache Hadoop. Shodan currently exposes data of several hundred of their servers (Shodan query: *product:namenode*). Moreover, Big Data developers can manage Hadoop clusters together with another Apache tool: Ambari. As of writing, Shodan is able to find just a few Apache Ambari servers (Shodan query: *ambari*), but some of them make use of the default administration credentials (a screenshot of the dashboard of one of such systems is shown in Figure A9 on the left).

There are also other Big Data tools that allow for visualizing easily large sets of data. For example, Kibana is commonly deployed together with Elasticsearch to visualize the collected information. Several thousands of open Kibana servers can be currently found with Shodan (query: *kibana content-length: 217*). As an example, Figure A9 shows on the right a screenshot of an open Kibana dashboard.

Finally, it is worth mentioning another Apache project that is increasingly being used by Big Data developers: Apache Spark. Such a tool provides a general-purpose cluster-computing framework for processing large amounts of data in parallel. Shodan can find several hundreds of monitoring panels of Apache Spark masters (Shodan query: *"Spark master" worker*) and workers (Shodan query: *Spark worker*). Figure A10 shows as examples screenshots of the panels of a Spark master and a worker exposed by Shodan.

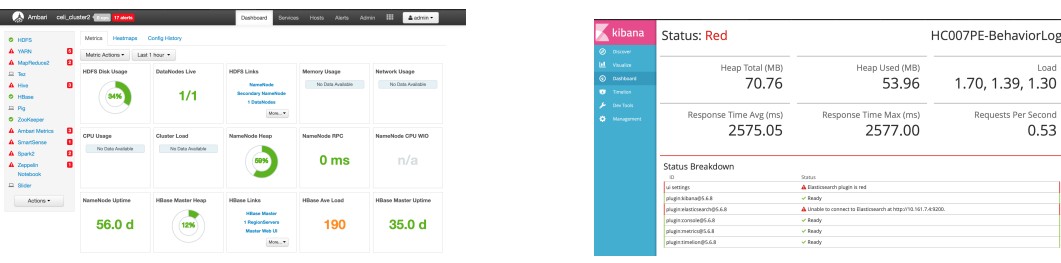

**Figure A9.** Apache Ambari (left) and Kibana (right) servers.

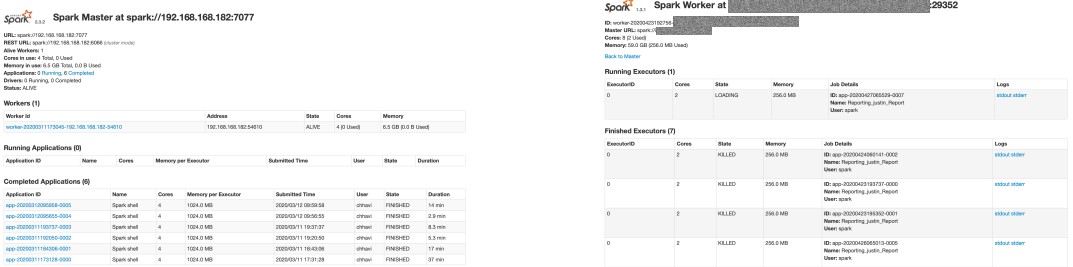

**Figure A10.** Monitoring panels of an Apache Spark master (left) and a worker (right).

*Appendix A.6. Cloud Computing*

Cloud computing has a relevant role in the Industry 4.0 paradigm since a significant amount of the processing is carried out by remote servers that centralize resources in a scalable way. Cloud servers usually run multiple virtual machines that are managed through a software like VMware vSphere. As of writing, Shodan detects almost 3000 VMware vSphere web clients (Shodan query: *Path=/vsphere-client/*). Moreover, Shodan currently detects almost 2000 running instances of VMware

vCloud Director (Shodan query: *X-VMWARE-VCLOUD-REQUEST-ID*), another VMware cloud infrastructure management software.

There are also other tools that run on the cloud that may be vulnerable to attacks. For instance, it is possible to find numerous Rsync servers, which are used to synchronize files between computers (Shodan query: *Product:rsyncd*). In addition, many software developers make use of automation tools that provide web dashboards with the shared projects. This is for instance the case of Jenkins: Figure A11 shows two open Jenkins dashboards discovered with Shodan (query: *port:8081 "Dashboard [Jenkins]"*). Another example is GitLab servers, which are typically used for software project development: more than 61,000 of them can be currently found through Shodan (Shodan query: *http.favicon.hash:1278323681*) and many of them allow for registering additional users to have direct access to the shared project code.

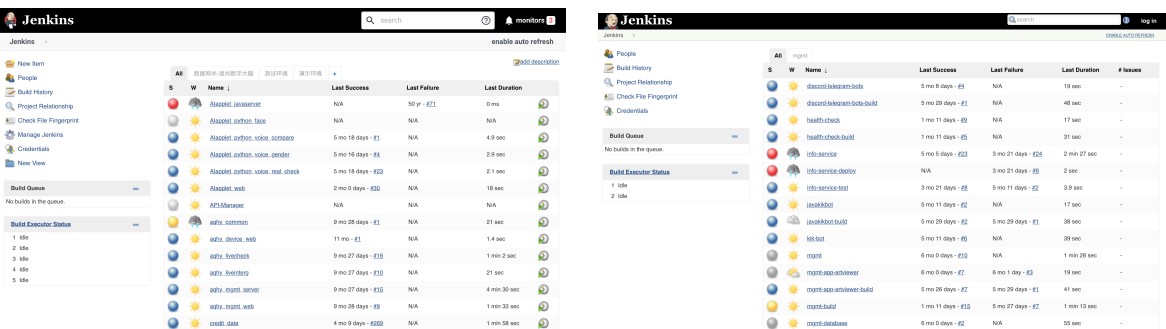

**Figure A11.** Examples of Jenkins dashboards found with Shodan.

*Appendix A.7. Blockchain and Distributed Ledger Technologies (DLTs)*

DLTs like blockchain provide decentralized transaction ledgers that are designed to avoid data tampering, thus implementing advanced cryptography mechanisms [92]. Thus, blockchain enables implementing decentralized applications that allow for exchanging transactions among entities that do not trust each other or do not want to incur in the additional costs derived from the involvement of certain middlemen (e.g., banks, distributors) [93]. Such features make blockchain appropriate for Industry 4.0 companies that require trusting their own employees and providers [16,94,95].

Most blockchain systems (e.g., Bitcoin [96], Ethereum [97]) are composed by regular nodes (i.e., nodes that only interact with the blockchain to read from it or to write data on it) and full nodes, which act like regular nodes but also validate the exchanged transactions. Such a validation process is often called mining, which involves a consensus mechanism and some sort of verification process. Thus, miners are nodes that validate collaboratively blocks of transactions before adding them to the blockchain.

Shodan is capable of finding blockchain miners. For instance, Antminer is one of the most popular mining platforms [98], since it provides dedicated and really fast hardware. As of writing, Shodan detects almost 1000 antminers (Shodan query: *antminer*). The transaction log of many Ethereum miners can also be discovered easily (Shodan query: *"Eth speed:"*), as well as hundreds of Bitcoin miners (Shodan query: *lastblock*). Furthermore, more than 3000 Ethereum clients are currently detected by Shodan (query: *"Ethereum RPC"*).

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
