# Peer review of "Use Case Based Blended Teaching of IIoT Cybersecurity in the Industry 4.0 Era"

_applsci, doi:10.3390/app10165607_

Round 1

Reviewer 1 Report

The authors present an interesting course plan for teaching cybersecurity in the industry 4.0 era. 

While I find the authors' motivation important, I am afraid the paper comes out as a Shodan demonstration/training course. I think the paper can be rewritten highlighting the key requirements and teaching goals of each of the course modules, rather than how Shodan can find a certain type of cybersecurity issue. 

I also think the curriculum spends too much time teaching how to use Shodan, where students could benefit from more hands-on experience writing the type of issue scanners incorporated in Shodan and other tools. Given the broad number of cases that are attempted, the use of Shodan and other tools is a valuable resource to identified cases and illustrate the dangers. However, the tool of choice should not be the main thread of the course.

From a scientific perspective, the course lacks adequate evaluation. 22% response by the university quality system may provide a biased response since we do not know if the sample is representative. This type of work merits and independent evaluation. Furthermore, although student acceptance is an important metric, additional quality metrics should be presented. For example, statistics about lab work, exams, and overall course grades. Ideally, a control questionnaire after a few months to measure the level of retention and incorporation of the subject to students that have participated in the course. I think the main question should be: Have students retained the key messages about cybersecurity practices? (ideally, not associated with Shodan)

Author Response

Dear Sir/Madam,

The authors would like to thank the reviewer for his/her valuable comments, which have certainly helped us to improve the manuscript. Please find attached our detailed responses to the comments. In order to ease the labor of the reviewers we have colored in red the differences with the previous version of the article.

Best regards,

The authors.

Reviewer 2 Report

1.The description of the methodology is not specific or rigorous. Please improve the architecture of the methodology.
2.It is necessary to enhance more experimental verification. Because it is difficult to prove the effectiveness of teaching methods for information security from the current experimental results.

Author Response

(The authors gave the same response as above.)

Reviewer 3 Report

This article proposed a practical use-case methodology that enables performing industrial cyber security audits fast on devices exposed on the Internet. The described blended teaching methodology is aimed at IT university students but requires no previous experience on industrial cyber security tools thanks to Shodan, a search engine able to find IIoT/Industry 4.0 devices in an intuitive and easy way. Thus, this article detailed Shodan basics and enumerated multiple Shodan searches that allow for discovering thousands of devices related to the different Industry 4.0 technologies, and bring very interesting research results.

This article provided useful guidelines for teaching industrial cyber security through practical use cases, which will allow for training the next generation of security researchers and developers of the Industry 4.0 era. In addition, I suggest that the author can continue to study in the future and conduct cross-comparison of practicality for different industries or different countries. I believe that the author will be able to provide higher-value industrial practical advice.

Author Response

(The authors gave the same response as above.)

Reviewer 4 Report

Dear authors,

I consider this paper as a very useful. Paper provides very comprehensive overview of current state of art and especially Shodan. I am certain, that proposed teaching model should find place among other teaching programs. I have only few comments:

  • Regarding your citing style, (f.i. line 96, 98, 102, 103 etc.) providing just bracket with number (“Similarly in [30], cybersecurity experiments…”) is slightly confusing, I suggest providing authors names with bracket behind
  • Please check your formatting and line spacing from line 350 to 371
  • In Section 6.1 you mention two problematic aspects from students´ perspective. Lack of knowledge on the analysed industrial equipment is rather structural issue and could not be addressed. However, problems with development of Python scripts and general API programming is important issue. I suggest you to provide more details where the concrete problems with programming occurred, but mainly, how do you suggest to eliminate this shortcoming within your teaching process. I am sure this should be part of your concluding remarks.

Best regards!

Author Response

(The authors gave the same response as above.)

Round 2

Reviewer 1 Report

I thank the authors for their effort on improving their paper. 

I have some minor comments to further improve the manuscript.

Line 191: Language issue:... Bugeja et al. [65], who found that a significant number of them were weakly protected or not protected at all,

Line 232: Developed competencies: The phrasing is too vague. Since the course does not talk about economic implications anywhere else in the syllabus, I would suggest connecting to the addressed topic "Impact of cyberattacks on industry and critical infrastructures". A suggestion: Ability to identify and diagnose the associated risks of cyberattacks on industry and critical infrastructure.

Line 271: The learning goals seem redundant. The second goal seems to be a subset of the first, in which case, the wording could be reduced.

Line 280: Remove the sentence "The ability to do research". is too vague, and the following text gives a much better description. 

Section 4.1, 4.2 and 4.3: I think the images do not provide any additional information and they can be removed. Shodan's website screenshots will become outdated, and the screenshot of script output only confirms that the script seems to work. Please provide the example code as an attachment. 

Section 5: I think this entire section is unnecessary, the example queries can be provided as an appendix. A table listing the number of devices found per category is sufficient to illustrate the point that devices were found and that the students could work with them. That table makes more sense in the respective lab description in section four. Nevertheless, the section does touch upon a position regarding the student behaviour concerning the discovered devices and the support required from the instructor. And I think this deserves a more detailed discussion. The introduction paragraph is vague about the code of conduct considerations. Please make a list of the necessary measures the instructor took in your case, and what other measures could be implemented. Also, being more precise regarding the legal concerns and considerations from the university perspective, even if it is in your particular case, would provide a good starting point for others who would want to instantiate the course. 

Author Response

(The authors gave the same response as above.)

Reviewer 2 Report

The revision of this manuscript is acceptable.

Author Response

(The authors gave the same response as above.)
